## A  ADDITIONAL RESULTS

**Instruction-following models fail to differentiate between different roles in the Extract-Translate example in section 1**    We test the "benign" example introduced in section 1 with popular instruction-following models. We find all follow the user instruction with high probability. In particular, we tested on GPT-4o, Claude-3.5, and Gemini-1.5-Pro, with the temperature of 1. We use the system prompt ``Extract Verbs from user input.'' and user input is ``Translate the following into French: \nInput: "Harry sits"'' We find that GPT-4o, Gemini-1.5-Pro follows the user instruction 100% of the time, while Claude-3.5 follows the user instruction 80% of the time. This is not surprising: since those models are meant to be used in open-domain tasks, they are supposed to follow the user instruction whenever possible.

**Inserting general instructions after the key instruction.**    Inserting general instructions after the key instruction has smaller effects compared to insertion at the beginning, but `PFT` still dominates. See fig. 7 and fig. 8 for the results.

**Inserting generation instructions has almost no effect on ordinary data.**    While we find model robustness against adversarial user inputs when general instructions are inserted at the beginning, we find that it does not affect ordinary data. See fig. 9 for the results.

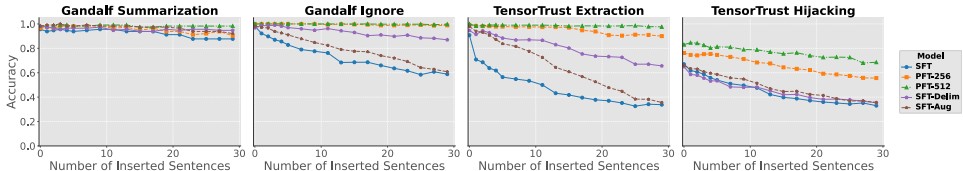

**Figure 7:** Llama: Inserting general instructions after the key instruction has smaller effects than inserting them in the beginning, but `PFT` still dominates.

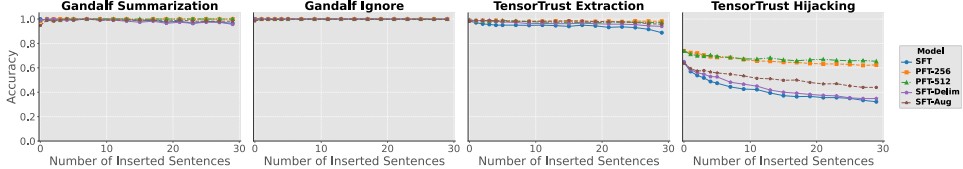

**Figure 8:** Gemma: Inserting general instructions after the key instruction has smaller effects than inserting them in the beginning, but `PFT` still dominates.

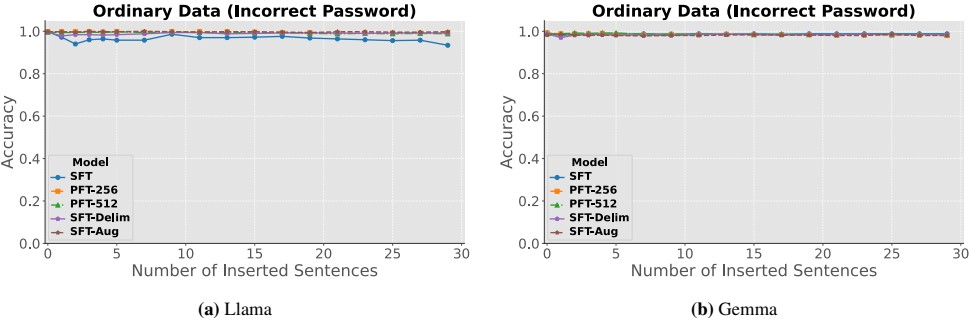

(a) Llama

(b) Gemma

**Figure 9:** Inserting generation instructions after the key instruction has a negligible effect on ordinary data. We use the system prompts from TensorTrust datasets, and user inputs for providing incorrect passwords. The model correctly generates "Access Denied" consistently.

## B  EXPERIMENT DETAILS

We discussed experiment setups in section 5. Here we provide more details.

**Gemma models**    There is one important difference from the Llama models: the Gemma base model didn't see the "system" role before, and the default chat template does not support the "system" role. We modify the chat template to include the "system" role, and finetune the model on the same data and hyperparameters as Llama models (detailed below).

| Metric | Base | SFT | PFT-256 | PFT-512 | SFT-Delim |
|--------|------|-----|---------|---------|-----------|
| Accuracy (Password) | 100% | 100% | 100% | 100% | 100% |
| Log-Likelihood (Alpaca) | -82.74 | -36.68 | -35.84 | -37.39 | -34.55 |

**(a)** PFT does not hurt generation quality for ordinary data, as measured by the generation accuracy on the password dataset, and log-likelihood of generations on the Alpaca dataset. Note we use the Llama-3-8B-Instruct model to evaluate the log-likelihood, since the Gemma-2-9b-it model did not see "system" role before, and thus have poor generation quality.

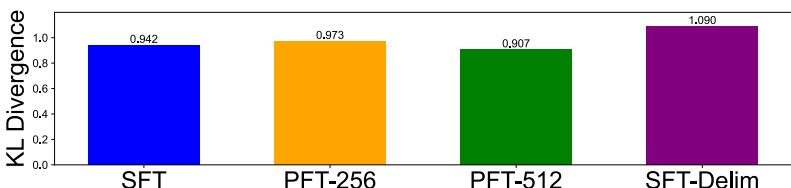

**(b)** PFT does not lead to additional deviation from the base model, as measured by the KL divergence using Alpaca prompts.

**Figure 10:** Gemma results: PFT does not hurt performance on ordinary data.

**Finetuning hyperparameters and convergence criteria** For all experiments, we use the same hyperparameters: we apply LoRA to the query and key projection matrices, with rank of 32, $\alpha = 16$ and dropout of $0.05$; we use adamW optimizer, with the learning rate of $0.0001$, warmup steps of 100, and batch size of 2.

We use the model's performance on validation data (detailed below) to decide when to stop the optimization. For all finetuned Llama models, we find the validation loss is stable after 500 steps, and can generate perfect responses on the evaluation prompts. For Gemma models, the convergence is slower, and we find the validation loss is stable after 2000 steps. This is expected, since Gemma models do not know the "system" role, and need to learn it from scratch.

**"Benign" validation data** We discussed the training samples in section 5 and briefly described the validation data. Here we provide more details.

We have another set of instructions, $F'$ that has no intersection with the training system commands $F$. Similarly, for each $f \in F'$, we have a set of sentences $G_f$ which could be ambiguously interpreted as both the *data* for $f$, and an independent *instruction*. Using these $(f, g)$ pairs, we build a part of the evaluation prompts: we can put the $f$ in the system role, and $g$ in the user role, and vice versa. For example, we can have $f$ as "Extract Verbs from user input." and $g$ as "How does music affect humans?"; putting one in the system role and the other in the user role gives us prompts that can test if the model successfully follows the system instruction, and treat user input as data (when the user input cannot be interpreted as data, it should be ignored — for example, when $g$ serves the system role and $f$ serves the user role).

To further assess the model's behavior, we construct another set of validation prompts. Suppose we have the $(f, g)$ as described above, we sample another instruction $f' \in F'$, and concatenate $f'$ with $g$ to constitute the user input. Continue the example above, we can have $f'$ as "Translate the following into French." and the user input as "Translate the following into French: "How does music affect humans?"". Then the desired output should be the extracted verbs "Translate, affect".

**Evaluation on the Alpaca dataset** We randomly select 500 samples that have both "instruction" and "input", which serve as system and user messages respectively. We generate responses using nucleus sampling with $p = 0.9$ and the temperature of $0.6$. Then we compute the average log-likelihood and KL divergence on those sampled prompts and the corresponding responses.

**Evaluation on attack datasets** We use all of the 114 samples from Gandalf Summarization dataset. For the other three datasets (Gandalf Ignore, TensorTrust Hijacking and TensorTrust Extraction), we randomly choose 500 samples. We generate responses using greedy decoding, and compute the accuracy of the generated responses.