# OpenReview forum: "PFT: Enhancing Prompt Injection Robustness via Position-Enhanced Finetuning"
_ICLR.cc/2025/Conference — Submitted to ICLR 2025_

### Official Review · Reviewer_HrGo · 2024-10-31

**Soundness:** 3
**Presentation:** 3
**Contribution:** 2
**Rating:** 5
**Confidence:** 4

**Summary:**

This paper addresses the vulnerability of Large Language Models (LLMs) in closed-domain applications, where distinguishing system instructions from user input is essential to prevent prompt injection attacks. The study reveals a limitation in Supervised Fine-Tuning (SFT), where models reliably follow system instructions only when these instructions appear immediately after the initial tokens. To address this, the authors introduce Position-Enhanced Fine-Tuning (PFT), which uses position IDs to better differentiate between system and user tokens.

**Strengths:**

1. The ideas are clearly presented, with contributions precisely defined.

2. Key terms and concepts are thoroughly defined, such as key instructions.

3. The writing is fluent, well-organized, and easy to read, with appropriate examples that enhance readability and understanding.

**Weaknesses:**

1. The contribution appears limited to "pure" closed-domain tasks, where users are restricted to providing only input data without additional task-specific instructions. In real-world scenarios (e.g., translation tools or platforms like ChatGPT and Copilot), users often add specific instructions, such as style preferences or formatting guidelines, which PFT does not address. Figure 2 suggests that PFT might not perform well in these more dynamic contexts.

2. The paper states that PFT imposes no performance penalty (Section 5.3), yet assumes that users will not add prompts, focusing solely on task input data. It would be helpful to analyze the trade-off between robustness and performance in a practical context where users provide detailed instructions. Examining the parameter \( d \) and its relationship with performance in such scenarios would clarify PFT’s effectiveness.

3. Including a comparison between PFT and a simple baseline prompt designed to direct LLMs toward system tasks would add value. Observers might find a "SOTA vs. SOTA+PFT" comparison more informative than "vanilla vs. vanilla+PFT."

4. In the robustness experiments (Figure 5), only data points with fewer than ten inserted sentences (\#ofSentences < 10) seem relevant for real-world usage. It would be helpful to highlight these cases more explicitly.

5. No code is provided, which limits the reproducibility of the findings.

6. Table 1 is difficult to interpret due to the lack of a legend or clear explanation of its components. Adding this information would improve clarity.

**Questions:**

1. In the performance experiments shown in Figure 3, was each model fine-tuned with its respective number of input sentences?

2. After the insertion process, do the system prompts remain fluent at the sentence level?

3. Including "prefixes" or "suffixes" in instructions for LLMs typically does not impact performance significantly. However, as shown in Figures 2 and 3, your experiments reveal a substantial drop in accuracy with just five sentences, causing a 50-80% decrease. Could you clarify the factors behind this observation?

4. Can I infer that "if the system prompt is sufficiently long, the model will remain robust and perform well even without PFT"?

---

> ### Author Response · Authors · 2024-11-18
>
> We are glad you find the paper clearly written. We release [code and data](https://anonymous.4open.science/r/pft-iclr-08E2), and we will update Table 1 for clarity.
>
> ***Regarding restriction to pure closed-domain settings***
> As we discussed in the general response above, allowing open-domain applications (i.e. allow user instructions) would blur the answers to our research question: when the LLM follows a user instruction when it’s not supposed to, it could be because (1) it doesn’t understand the distinction between system and user roles, or (2) it doesn’t understand that such instruction from the user is not allowed. On the other hand, closed domain settings also have their applications. For example, we might ask the LLM to decide whether to recommend an applicant based on the resume (passed in as user input).
>
> About Fig 2 implies PFT might fail in more dynamic contexts: Fig 2 only shows the problem in SFT when extra instructions are added to the system prompt (not user prompt), so we can’t infer the performance of PFT in more dynamic settings.
>
> ***Regarding “SOTA+PFT vs SOTA”***
> As argued in the general response, we did try our best to use elements from SOTA methods (e.g. Instruction Hierarchy and StrucQ). However, to fully implement the SOTAs, we can’t perform the controlled experiments to answer our research question: almost all those works permit some “allowed” user instructions, which we argue doesn’t help elucidate our main points.
>
> ***About Figure 3***
> The finetuning data (described in 5.1) has system prompts that only include the key instruction. We stress-test the LLM in the OOD setting in Fig 3 to show its fragility. After insertion, the model output still remains fluent.
>
> ***About the impact of prefixes and suffixes***
> We agree that the big performance drop is shocking. In Sec 3, we investigated this phenomenon and hypothesized that the LLM learns the shortcut of following the first sentence after the beginning of the sentence during security fine-tuning (probably because of attention mechanisms).
>
> ***About “if the system prompt is sufficiently long, the model will remain robust and perform well even without PFT”***
> Do you mean the model would remain robust if we use very long system prompts during security finetuning? We actually test this with SFT-Aug, by augmenting the length of the system prompt. We find it does help, but when the evaluation system prompt is longer than those used in training, the performance still drops way more than the PFT-tuned model.

---

> > ### Comment · Reviewer_HrGo · 2024-11-23
> > **Response to the rebuttals**
> >
> > Thank you for your rebuttals. While the research approach is impressive, my main concern is the limited scope of the scenarios. As you mentioned, most SOTA works commonly allow user instructions, which leaves me uncertain about fully appreciating the research value of this work. Thus, I will maintain my score.

---

### Official Review · Reviewer_JdUj · 2024-11-02

**Soundness:** 2
**Presentation:** 2
**Contribution:** 2
**Rating:** 3
**Confidence:** 5

**Summary:**

This paper has two main contributions: (1) They demonstrate that LLMs are more likely to follow instructions *closer* to the beginning of the input. This is quite a surprising and interesting phenomenon. (2) Based on this observation, the authors suggest a modified SFT procedure to combat prompt injection attacks. This defense works by shifting the user input by a constant offset.

**Strengths:**

1. The observation regarding how position of an instruction affects the model’s LLM to follow it is scientifically interesting. The experiments are convincing for this point. It is very interesting to see that the attack token logit shoots up with only a small number of inserted sentences and then plateau.
2. The proposed defense is simple and seems to work at least in a limited setting.

**Weaknesses:**

1. **Choice of the model to be fine-tuned.**
    1. Why do you fine-tune instructed models instead of the base pre-trained model? The instructed models can already solve these tasks so it is perhaps expected that the utility is not hurt via PFT. These models are also already aligned (via RLHF or safety tuning) so they should already be somewhat resilient to such attacks.
    2. Or, the authors intend to propose PFT as an additional step after RLHF? However, I believe that
    the authors are suggesting to replace SFT with PFT, and if so, the base model for this experiment should be the base Llama-3-8B, not instruct.
2. **Weak defense baselines.**
    1. I believe that this experiment is missing two important baselines: Instruction Hierarchy and StruQ. My understanding is that the authors intend to use “Delimiter-enhanced SFT” to represent StruQ and “Data-augmented SFT” to Instruction Hierarchy. However, I would suggest reproducing these baselines exactly or as close as possible and compare to them in the same setup (same dataset). Without this direct comparison, it is impossible to conclude whether PFT is better.
    2. Data-augmented SFT: What happens if the augmentation is done to the user prompt instead of the
    system prompt? This would be more like StruQ.
    3. Data augmentation is used in both Instruction Hierarchy and StruQ, and it is different from what’s done in this paper. What about training against a subset of the attack? If we do that and combine with PFT, is there any improvement?
3. **Weak attack baselines.** The authors should consider evaluating against stronger attacks, e.g., a set of handcrafted attacks from StruQ, jailbreaks, or even automated attacks such as [PAL](https://arxiv.org/abs/2310.08419), [TAP](https://arxiv.org/abs/2312.02119), [AutoDAN](https://arxiv.org/abs/2310.04451), or [GCG](https://arxiv.org/pdf/2307.15043). This would help with comparison to the prior works.
4. **KL divergence metric.** It is mentioned that the KL divergence is computed between p_model(output text|prompt) of the model before and after fine-tuning. As far as I know, there is no way to efficiently compute this because the set of “output texts” is just too large. My guess is that the authors compute the KL divergence at *each* token conditioned on the prior tokens, which is a different quantity from what is stated.
5. **Not applicable to real-world / more diverse use cases.** This defense does not hurt the utility because all of the Alpaca samples put the instruction close to the beginning of the input. This defense would fail immediately when legitimate instructions are part of the user input and placed anywhere, e.g., in chatbots.

### Nitpicky Stuff

- In Table 1, calling the model before fine-tuning a “base” model is slightly confusing. In my understanding, this is an aligned and instructed model, but "base model" often refers to a pre-trained model on the next-token prediction task. I might suggest simply calling it "before fine-tuned," or at least make it very in the text or caption (not delay until Section 5).
- L193 (”this is not a strong enough signal for LLMs…”): I believe that this statement is plausible but still not very convincing. Could you explain more why the delimiters would be a weaker signal than the begin-of-text token? Is it because there is only one begin-of-text token in every sample, and it is always at the beginning? -- I see that this is discussed later on in Section 3.2. It might be a good idea to refer to it here.
- L211 (”Also the first insertions have a dramatic effect…”): I can understand the message from looking at Figure 3, but this statement is hard to follow and is pretty confusing to me.
- L285 (”assigning a unique signature to each token based on its role (system or user), we create a continuous, fine-grained distinction throughout the entire input.”): I believe that this sentence is not an accurate description of PFT. Each token is pushed back by the same offset $d$ which does not sound like a unique signature to me. I’m also not sure what “continuous, fine-grained distinction” refers to.

**Questions:**

1. Table 1: I'm also quite surprised that the attacks are very effective on these "base" safety-aligned models. Is there any detail I can see regarding the implementation? For example, what is the system prompt for these models? Did you place the main instruction from these datasets in the system prompt or the first user message? Llama-3-Instruct has no system prompt by default and likely expect instruction to be in the first user message. Violating this format might hurt the model's performance.
2. Which model is used for this experiment? Is it the same model from Section 2.1, or a different public model? I think this information bit is rather important, i.e., the result is very much expected if it is the model from Section 2.1.
3. L358 (”We ask GPT-4 to filter out cases where the model still misinterprets the user input as a separate task.”): Could you explain more why this step is necessary?
4. Figure 6(a): Is the log-likelihood computed on generations or on the correct answer? The paper says
”generations” but this does not make sense (e.g., confident wrong answer)?

---

> ### Author Response · Authors · 2024-11-18
>
> We are glad you find our observation scientifically interesting, and that the experiment is convincing for this point.
>
> **Regarding why we use the fine-tuned models instead of the pre-trained models as the starting point**: Without instruction-tuning, the LLMs can’t be used as bots since they don’t follow instructions, so there is no point in studying the security of those models. You point out that maybe safety finetuning might make the models robust to attacks, but we find the exact opposite: the instruction-tuned model has almost no resilience to the attacks (Table 1). We think it is a result of instruction-tuning: the model is taught to follow any instructions no matter where they appear.  We emphasize that safety fine-tuning you mentioned (teach model not to say dangerous stuff etc) is very different from the security finetuning we performed in later stages (teach model to treat user input as data).
>
> ***About the role of SFT/RLHF/PFT***
> There seems to be some confusion here. The SFT in this paper is security finetuning, which is described in section 2. This is performed on the instruction-tuned models (Llama-3-8B-Instruct and Gemma-2-9b-it).  We noticed that confusion might arise from our use of “SFT-tuning”. We will replace it with “security-supervised-finetuning” for clarity.
>
> If we resolve this confusion, then it’s clear that PFT is replacing the standard “security-supervised-finetuning” i.e. using the same dataset and training objective, but performing additional positional ID manipulation.
>
> ***About reproducing Instruction Hierarchy and StruQ***
> The experiments as presented are our best effort at reproducing the two works for the research question in this paper: our construction of training data follows the procedure described in the Instruction hierarchy, and we augment delimiter tokens in StrucQ. The other parts of the two works are not relevant here. Instruction hierarchy additionally includes a dataset for open-domain applications, and strucQ include secure front end for Completion attacks. But our focus here is only on the closed domain, and we don’t include Completion attacks in the attack data (attackers do not have access to the delimiter token).
>
> ***About weak attack baselines and adding attack data during training***
>
> The goal of this paper is to model robustness w.r.t. different system prompt designs. As we said in the general response, the current “weak” attack baseline, actually shows a generalizable defense (since we only use benign data in training). Adding more advanced attacks while adding actual attack data during training doesn’t help clarify the focused question here.
>
> ***About KL divergence***
> We measured the Kl between the two conditional distributions (line 419). Indeed, the estimation of this quantity has a big variance, but this is what people mostly reported in the alignment field (e.g. https://proceedings.mlr.press/v202/gao23h/gao23h.pdf ).
>
> ***About calling the model before fine-tuning a “base” model***
> We agree this seems to cause confusion here (especially about the role of SFT/PFT)! We can call it “before security-finetuned”.
>
> ***About assigning a unique signature to each token based on its role***
> After the proposed position id manipulations, each user token has much larger position ids than the system tokens. We consider this a token-wise signature.
>
>
> ***About the results in table 1***
> We include the code and dataset. For table 1, each system prompt just contains the key instruction and nothing else. We think it’s not surprising because “safety-tuning” doesn’t seem to directly cope with security issues (see our response to the first point). Also, Llama-3-Instruct does have system prompt by default.
>
> ***About L358***
> When we construct the desirable model responses, we ask the Llama-3-Instruct to answer by treating user input as data. In some rare cases, the model fails to do that. We use GPT4 to filter out such cases.
>
> ***About figure 6(a)***
> As stated in the caption, the accuracy is on the password dataset, and the loglikelihood is on the Alpaca dataset.

---

> > ### Comment · Reviewer_JdUj · 2024-11-22
> >
> > Thank you for the response!
> >
> > **SFT vs PFT**
> >
> > > Without instruction-tuning, the LLMs can’t be used as bots since they don’t follow instructions, so there is no point in studying the security of those models.
> >
> > I believe that prior work like StruQ or Instruction Hierarchy improve the security of SFT, i.e., they propose an enhanced SFT which is applied to the pre-trained model. However, in this paper, you're proposing another fine-tuning round on already instructed model (e.g., model that has gone through a non-secure SFT). I think it's a good idea to mention this clearly in the paper, and there're some pros and cons worth discussing between the two approaches.
> >
> > **Comparison to StruQ/ Instruction Hierarchy and evaluating against stronger attacks.**
> >
> > While I understand the authors' reasoning for not directly comparing to the original StruQ and Instruction Hierarchy and not using the stronger attacks, I believe that it unfortunately limits the contribution of this work.
> >
> > **About KL divergence**
> >
> > Thank you for the clarification and pointing out the related work!
> >
> > **Other points**
> >
> > > We include the code and dataset.
> >
> > I can't seem to find any link to the code in the main paper or the appendix. Would the authors mind pointing this out to me?
> >
> > > Also, Llama-3-Instruct does have system prompt by default.
> >
> > This is pretty minor point, but I still don't think this is the case. According to the [system card](https://www.llama.com/docs/model-cards-and-prompt-formats/meta-llama-3/) and [HF tokenizer config](https://huggingface.co/meta-llama/Meta-Llama-3-8B-Instruct/blob/main/tokenizer_config.json), there's no predefined system prompt for Llama-3-Instruct. On the other hand, Llama-3.1-Instruct does have some default system prompt (from [HF config](https://huggingface.co/meta-llama/Llama-3.1-8B-Instruct/blob/main/tokenizer_config.json)). All that said, it doesn't really matter in your setup as it seems like you overwrite the system prompt with a custom instruction.
> >
> > However, I believe that including the key instruction as the system message is a bit non-standard for chat models. If the authors focus on instruction-following models (non-chat, like Alpaca), then there's no system message. If the authors want to work on chat models, then instruction should be under user tag and not system. This is overall a minor point, but to me, it is more (unnecessarily) toy-ish and makes it less applicable to the real-world use cases.

---

### Official Review · Reviewer_wZvs · 2024-11-04

**Soundness:** 2
**Presentation:** 2
**Contribution:** 2
**Rating:** 3
**Confidence:** 3

**Summary:**

This paper proposes PFT, a novel position-enhanced fine-tuning approach that leverages position IDs to more effectively distinguish between system and user tokens. The experimental results demonstrate that PFT improves the robustness of SFT-tuned models against prompt injection attacks, even when the key instruction is placed arbitrarily in the system prompt, without compromising performance.

**Strengths:**

This paper explores the security and robustness of models under different conditions of prompt structure. It finds that SFT-tuned models, which are secure when key instructions are positioned at the beginning of the prompt, become vulnerable when these instructions are placed later. The study demonstrates that the proximity of the key instruction to the start of the input significantly impacts the model's adherence to the designated task. To address this vulnerability, the paper introduces Position-Enhanced Fine-Tuning (PFT), a method designed to protect models from adversarial inputs by ensuring robustness and maintaining performance, regardless of where instructions are positioned within the prompt.

**Weaknesses:**

1. The definition of the problem lacks clarity. Specifically, the formal definition of a "key instruction" in system prompts is ambiguous. How and why does it differ from other system prompts? It is challenging to distinguish key instructions from other contextual prompts,  especially since other prompts can sometimes serve as the context for the key instructions. This will lead to complex scenarios that needs a detailed discussion.
2. Following the first point, the generalizability of PFT needs to be clarified, as key instructions vary across different contexts. The validation dataset used in the paper mirrors the examples provided in the introduction, which does not suffice to demonstrate PFT's applicability in more complex and practical scenarios.
3. Stronger attacks are necessary. In the realm of prompt injection, several existing studies employ learning-based methods to launch attacks, as referenced in [1]. I suggest that the authors include experiments to test the resilience of PFT against these types of attacks.
4. The paper lacks a discussion on adaptive attacks. If attackers know the PFT method and can fine-tune the model accordingly – for instance, tuning it to adhere strictly to user instructions?

[1] Pasquini, Dario, Martin Strohmeier, and Carmela Troncoso. "Neural Exec: Learning (and Learning from) Execution Triggers for Prompt Injection Attacks." arXiv preprint arXiv:2403.03792 (2024).

**Questions:**

See above.

---

> ### Author Response · Authors · 2024-11-18
>
> Thank you for your reviews. Here are our responses to your questions:
>
> **The definition of “key instruction”** We defined it in the second paragraph of the introduction (Line 43). The contextual prompt should be treated as part of the key instruction.
>
> **About “The validation dataset used in the paper mirrors the examples provided in the introduction, which does not suffice to demonstrate PFT's applicability in more complex and practical scenarios.”** We feel there is a misunderstanding here. The validation dataset (described in paragraph of 346) is for in-domain validation and is indeed similar to the benign examples in the introduction.  However, the main evaluation results reported are on adversarial inputs, are very different, and are not used during training or validation.
>
> **Regarding advanced adversarial techniques**: as we stated in the general response, the current setup (training with benign data and evaluation using attack diverse data) suffices to demonstrate the model’s generalizable defense against injection attacks. This metric provides sufficient coverage and reflects the model's sensitivity to system prompt design.

---

### Official Review · Reviewer_PTRg · 2024-11-04

**Soundness:** 3
**Presentation:** 2
**Contribution:** 2
**Rating:** 6
**Confidence:** 3

**Summary:**

By adapting the position-dependent encoding,
it is possible to strengthen LLM's ability to
follow system prompts and reject prompt
injection style attacks.

**Strengths:**

The paper identifies shortcomings of existing
models, presents experiments to demonstrate these
shortcomings, and then presents a simple and
elegant solution.

From the paper's results, it appears that the
proposed mechanism does improve a certain form
of robustness.

I think the approach might be of interest to
researchers.

**Weaknesses:**

Overall, I struggled to evaluate this paper.
The paper has some interesting results, and I'm
not sure what to make of them.  So I'm not sure
whether to recommend acceptance or not for this
paper.

The paper's notion of robustness is: the LLM
should be able to tolerate irrelevant instructions
appearing before the key instruction, in the
system prompt.  Is this important?  Do real
applications use system prompts that contain many
irrelevant instructions before the key instruction?
I'm not sure.  Therefore, I'm unsure whether the
problem this paper tackles is important.
I encourage the authors to provide examples
or evidence of real-world applications where
this property is important.

The notion of robustness that I'm used to is a
bit different: can the LLM resist all attacks?
If we pick some class of attacks, what is the
attack success rate of the strongest attack in
that class?  In other words, increasing robustness
means reducing the attack success rate of some
attack -- and this is evaluated for average-case
prompts (i.e., ones that will appear in real
applications), rather than worst-case prompts
(e.g., where we add extraneous instructions at
the start).  That's not the notion this paper
takes on, though.

The paper starts from a premise about how LLMs
will/should be used (put the instruction in system
message, the data in user message), a premise that
I am skeptical about.  Then it draws some conclusions
about that usage.  I'm not sure whether those
conclusions generalize to ways of using LLMs that
I think are more appropriate and more common.
Also, I'm not sure whether the paper's results
generalize to multiple models (different LLMs).

I also struggle to tell whether this paper's
results are primarily telling us something about
prompt injection (inserting malicious instructions
into a field that is only supposed to contain
data) or system-message-following (supplying user
messages that contradict/violate rules/guardrails
established in the system message).  Details on
experiments are vague and so it's hard for me to
tell what is being tested.


Detailed feedback:

Abstract: I had a hard time understanding the abstract.
I don't understand what is meant by "the key instruction",
or what system instructions have to do with prompt
injection.

Sec 1, paragraph 2: I don't agree with the claim here
about how engineers typically build systems with LLMs.
I don't think the zero-shot prompt is typically put in
the system instructions.  Instead, I think the prompt
or instruction is typically put in the user message,
and the system message typically contains guardrails
that constrain what types of prompts/instructions will
be followed.  If you disagree, I encourage you to look
for quantitative evidence (perhaps surveying some
collection of systems built by others).

Sec 1: the paper seems to conflate two issues that I
consider separate: (a) prioritizing system instructions
over user instructions when they conflict; (b) ignoring
all instructions in the data part of the user message.
I consider prompt injection to be problem (b), and
problem (a) to be a separate problem.  The introduction
does not distinguish between these two, and that makes
it harder for me to understand what problem the paper
is and isn't solving.

Or, to put it another way, we can distinguish between
system instructions, user instructions, and user data.
The paper does not seem to clearly distinguish between
these.  It seems to assume the system message contains
instructions and the user message contains data.  In
my experience, that is not representative of how LLMs
are used in real systems and not representative of how
production LLMs are trained in practice.  My experience
is that the system message (if present) contains
instructions (such as guardrails or restrictions or a
definition of the domain/scope), and the user message
often contains a combination of both instructions and
data.

Table 1: It's not clear to me what this is showing.
What does a number like 10% mean?  What is "Gandalf
Summarization"?  I think the table caption should
provide a self-contained explanation, or the table
should be moved to later in the paper after all key
concepts have been defined.  I think the table
caption should specify what the number/percentages
mean (what are they measuring?  attack success rate?).

Sec 2.1: So many details are missing.  What base
model do you use?  How large is your dataset?  What
are the attacks you evaluate against?

I am quite skeptical of the claim that the model
is secure.  I don't believe you've evaluated against
enough attacks to draw such a conclusion.  There
are some quite strong attacks, such as GCG and TAP
(modified to create prompt injection attacks), which
are not considered here.  Therefore, it is not
warranted to conclude that the model is secure, as
you don't know whether it will be secure against
these more sophisticated attacks.

Sec 2.2: Does not match my experience.  In my
experience, general instructions ("You are an
AI assistant") go in system messages, the
"key instruction" (e.g., "Translate this to
French") goes in the user message, and background
knowledge and context and few-shot examples and
RAG-retrieved excerpts typically go in the
user message.

Many details are missing.  What model did you
use?  What attack did you use?  What was the
dataset and tasks for evaluation?  What is meant
by an "attack dataset"?

Fig 2: How is accuracy measured?  How do you tell
whether the model's respons is accurate?

Sec 3.1: It might help if you stated what
prompt injection attack techniques you used.
From Fig. 4, it sounds like maybe you used
the most naive attack, just directly asking
the model to do something different.

Fig 4: I'm not convinced the model is trained
to know what "user's input" means, so I'm not
sure whether this is a fair test of LLM capabilities
or if this is the right way to use LLMs for this
kind of task.

Sec 4: Do you have any explanation whether
it is better to have a fixed-length gap between
system vs user message (i.e., user message
starts at token $k+1+d$) or have the user
message start at a fixed position (i.e.,
user message starts at token $d+1$)?  Have
you tried both?

Sec 4: It seems this assumes that messages
will always appear in the order: system
message first, then user message second.
In contrast, existing LLMs allow them to be
interspersed arbitrarily.  Does this restriction
cause any loss in flexibility?  Does it matter?

Also, how do you propose to handle multi-turn
interactions?  How will they be encoded, and
where will gaps appear or not appear?

Sec 5.1: Are you doing SFT on a model that
has already been instruction-tuned, or on a
base model that has not?  It sounds like you
are doing SFT on a model that was already
instruction-tuned?

Sec 5.1: How do you generate the desired
responses to pairs where the system prompt
and user input have been swapped?

Sec 5.1: Will you make your dataset and
code available, to support reproducibility?

Sec 5.2: I don't understand what the attacks
are.  Please provide more detail.  For
Gandalf Summarization, the paper cites
Lakera AI, 2023b, but that reference is
bogus: the URL is for the Lakera main web
site, which clearly does not have the claimed
information, nor could I find any other
webpage on the Lakera web site with the
listed title.  The same comments apply
to Lakera 2023a.  Please provide direct
links or references for each attack, and
preferably describe the attack in a
self-contained way in the paper.

I suspect you might not be testing against
the strongest prompt injection attacks,
such as found in other papers on the
subject.  For instance, I would be more
convinced if you had evaluated against
completion attacks, TAP attacks, and GCG
attacks.

Sec 5.3: I disagree with using "most
robust" here.  I believe what you actually
measure is the property "isn't distracted
by extraneous instructions at the start".
That isn't the same thing.  What you study
is one narrow, specific aspect of robustness.
This is relevant if real systems use
system prompts that start with a lot of
generic, extraneous instructions.  It is
less clear how relevant it might be if
real systems don't do that.  And it does
not necessarily imply that PFT can defend
against stronger attacks.

Fig. 6: This is missing a key metric: you
should also show the accuracy and
log-likelihood for the undefended base model
with none of these defenses applied.  That's
what users will really care about: does the
defense harm the utility of the model,
compared to existing models with no defense
applied; not whether your defense is about
as good as other plausible defenses.

Consequently, I don't think the paper can
reasonably claim that PFT doesn't hurt model
performance.  It might be true, but I don't
think the paper has evaluated that.

Fig. 6(b) shows that there is some deviation
from the base model.  Is 0.5 KL divergence
a large deviation, or a small one?  It's hard
to know.  Perhaps looking at a small random
sample of responses from both the base
model and PFT model would help.

Sec 6: The paper seems to claim that OpenAI's
instruction hierarchy method has fragility
when the key instruction appears later in
the input.  But it's not clear that this
has actually been tested.  I would find this
analysis of the instruction hierarchy method
more compelling if the paper empirically
measured this, e.g., on GPT 4o-mini.

The paper appears to claim that StruQ has
fragility in this case as well, and that PFT
is more robust.  However, I don't think this
has been demonstrated, as the paper does not
evaluate StruQ.  StruQ uses some techniques
that were not tried in any of the models
evaluated in this paper.  I think the comparison
to StruQ would be more convincing if the paper
compared to a StruQ-trained model.  I don't
think we should view StruQ as designed just
to defend against Completion attacks; it seems
like it is trying to handle all prompt
injection attacks, as much as possible.

The paper seems to be missing one piece
of related work, BIPIA (Yi et al,
arXiv:2312.14197).

**Questions:**

How does the robustness against prompt injection
attack of your model compare to prior work, such
as StruQ's and BIPIA's models?  How robust is
it, when evaluating on the strongest attacks,
e.g., Completion attacks, TAP, and GCG?

How do your results change if you put both the
instruction and the data in the user message?

What is the accuracy and log-likelihood
(Fig. 6(a)) for the undefended base model?
How do I interpret a KL divergence of 0.5?

How common is it for real prompts to contain
extraneous instructions before the key instruction?

How should I interpret the paper's results
(see discussion above)?

---

> ### Author Response · Authors · 2024-11-18
>
> Thank you for your careful and engaged reviews! We released the [code and data](https://anonymous.4open.science/r/pft-iclr-08E2), as well as added clarifications to some of the confusion in the general response.
>
> ***Regarding whether the research question is important***
>
> We find the failure example in Figure 2 to be surprising and illustrative of some general phenomenon: simply adding “You are an AI assistant” before the key instruction is enough to subvert the defense setup during security finetuning. This poses a surprising challenge for practical use: the system might become fragile when developers place some general instructions/guardrails along with the key instructions. An ideal secure system should remain robust to those extra sentences.
>
> ***About the notion of robustness***
>
> We agree our notion is different from the popular one, and we argue our notion is also an important aspect of the robustness that not many papers look at. We included a detailed explanation in the general response. We will make it clearer in the introduction.
>
> ***About how the LLM should be used and the confusion about the results***
>
> In popular platforms like GPT-store and Coze-store, developers control the model's functionality through the system instruction. Here are screenshots for example usages in the two platforms ([GPT store](https://anonymous.4open.science/r/pft-iclr-08E2/additional_figs/gpt.png) and [Coze Store](https://anonymous.4open.science/r/pft-iclr-08E2/additional_figs/coze.png)). Also, our results are validated on both Llama-3-8B-Instruct and Gemma-2-9b-it.
>
>
>
> ***About whether PFT hurts the model utility***
> We want to eliminate the concerns that our position id manipulation leads to worse utility compared to standard SFT (since they only differ in the position ids). Our results compare their performance. We didn’t put the performance of the undefended model in the main text since the goal is to compare it with SFT-ed models. However, in the appendix, we show the undefended model results (Fig 9), which is similar to those after security finetuning.
>
> ***Regarding Detailed Feedback***
>
> - About the abstract: we agree it’s not immediately clear what the key instruction means in the abstract, but we define it soon in the second paragraph of the introduction (line 42)
> - On “the paper seems to conflate two issues that I consider separate”: since we consider the closed-domain bots where the user input should be completely treated as data, we think (a) and (b) are the same thing.
> - About Table 1: numbers like 10% mean the “percentage of times the model
> generates the correct response” (line 406), and "Gandalf Summarization" refers to the dataset in https://huggingface.co/datasets/Lakera/gandalf_summarization. In the caption, we cross-reference to details in section 5.2. But we will improve the captions to make the interpretations clearer.
> - About the model and data details: for demonstration purposes, we defer the model and dataset description in section 5.1.  See https://anonymous.4open.science/status/pft-iclr-08E2 for details.
> - About whether it’s better to use a fixed position: this is an interesting alternative. However, it won’t work when the system prompt is longer than the pre-chosen fixed position d+1. Using a fixed-length gap has no such problems.
> - About the order of system message and user message: here we always have system message appear before the user message. For the reverse order to work, we need to finetune the LLM with the reverse order.
> - About multi-turn interactions: this is a good question and an interesting direction for future work. We briefly discussed it in Line 525: we need more complicated methods to delineate the roles and dialogues.
> - About more advanced attacks: indeed the finetuned model might fail at more advanced attacks. But this is not the focus of this paper: we want to assess model robustness under different system prompt designs. We think the current diverse attack datasets are enough for this purpose. See the general response for more detailed discussions.
> - About comparison with other works.  As we argue in our general response, the goal is not to say PFT is superior to any of the related work but to provide evidence that it’s important to differentiate tokens from different roles. The current setup gives us evidence through controlled experiments. However, we did try our best to include relevant components from the related work. We cannot incorporate other parts as they allow the model to follow certain user instructions, violating the settings considered here.

---

> > ### Comment · Reviewer_PTRg · 2024-11-21
> > **On the problem setup, obeying system messages vs prompt injection, and closed domain vs open domain**
> >
> > I remain unconvinced about certain aspects of the problem setup, or the language used to describe it.
> >
> > I believe the paper is using the wrong definition of closed-domain.  Many GPTs on the GPT store have a narrow domain (e.g., "You are a personal trainer, and you can help devise an exercise plan"), but users interact with them via chat and instructions ("Suggest an exercise plan to prepare for a half-marathon.").  The authors seem to assume that closed-domain bots do not use chat or instructions and are only fed data, but I believe that in practice most closed-domain bots are designed to be interacted with by having the user type chat or instructions, and rely on an instruction-tuned model.  As such, those GPTs are not a good match for the kind of experiments done in this paper.  In particular, in those GPTs, the primary requirement is that the system messages be followed (in the event of a conflict between the user message and system message, the instructions in the system message should take precedence), not that the model
> > avoid following instructions in the user message.  To put it another way, in those GPTs, the risk is not one of prompt injection, but rather of violation of system-message-following.  So the paper's experiments are not relevant to the primary security requirements for those closed-domain bots.
> >
> > The paper wrongly conflates closed-domain bots with prompt injection and open-domain bots with system-message following.  However, system-message following is a key requirement for both closed-domain bots and open-domain bots (as such bots typically put guardrails and other rules in the system prompt, and must handle arbitrary user chat and instructions in the user message), but prompt injection is not very relevant.  Resistance to prompt injection is a key requirement for LLM-integrated applications, but system-message following might be less relevant (because those applications typically put a zero-shot prompt in the user message but do not use any system message).
> >
> > Put another way, the authors seem to assume that in closed-domain bots, the user message should be treated purely as data, and no instructions should be interpreted or followed.  As I hope the example above illustrates, I believe this is a faulty premise.  With a personal trainer bot, typical user messages might be "Suggest an exercise plan.." or "Should I use Madcow 5x5 or Strong Lifts 5x5?", which involve instructions or chat.  Or, in the example bots listed in the author response, typical user messages involve instructions or chat, e.g., "Extract the verbs from 'He has...'", "What are the verbs ...?"  See also review HrGo weaknesses 1-2.  Allowing user instructions does not make the bot into an open-domain bot; it still has a limited domain (e.g., physical trainer; verb extraction).
> >
> > If the authors are most interested in prompt injections, I think the authors should focus on LLM-integrated applications, where the application is written in traditional code and calls the LLM API to perform certain subtasks (typically, using zero-shot or few-shot prompting).  In these settings, the variable input is indeed intended as input, not instructions, which is a good match for the paper's experiments.  But in that setting, they are normally not implemented by putting the prompt in the system message and the data in the user message.  Instead, they are typically implemented by concatenating the prompt and the data (perhaps separated by a blank line or separator or delimiter), and putting all of that into the user message.  Thus, the paper's assumption that the prompt will be in the system message and the data in the user message does not match how most applications currently use LLMs.
> >
> > I want to put these complaints into perspective.  I do think it is interesting to discover that instructions that appear near the beginning of the system message are more resistant to prompt injection than those that appear later.  I also think the authors proposed intervention / defense is interesting.  I think both of these are valuable contributions and new insights for the field.  I think those positive contributions outweigh my complaints about some of the positioning and language.  I care about this mainly because, if published, this paper will set a precedent and framework that may influence future work in the community.

---

> > ### Comment · Reviewer_PTRg · 2024-11-21
> > **Other miscellaneous, secondary remarks**
> >
> > I think that you will be able to draw more convincing conclusions if you evaluate against stronger prompt injection attacks.  We care about how secure the model is against prompt injection attacks.  Since we are dealing with an adversary, this means we care about how secure the model is, against a worst-case attack.  i.e., we care about the max attack success rate, taking a max over all attack strategies.  Therefore, if you want to draw conclusions about security or robustness against prompt injection, I think it's fairly important to consider the strongest attacks.  If there is an adversary who is trying to attack our systems, it is unclear why they would limit themselves to avoid the strongest attacks.
> >
> > I continue to argue that the paper does not compare to StruQ, and it should.  The paper contains statements that I think could leave a misleading impression that it has compared to StruQ, but I do not believe it actually has.  Doing experiments with special delimiter tokens is not the same as comparing to StruQ.  StruQ involves more than just use special delimiter tokens; and moreover the specific details of how they are used and trained makes a significant difference to their effectiveness.  For that reason, I encourage you to compare directly to a StruQ model, as released by the authors of StruQ, or to revise the statements made about StruQ.  I think you can do better than what is in the paper.  I don't understand why the author response says you can't perform controlled experiments with StruQ.  StruQ doesn't permit some allowed user instructions to appear in the data field.
> >
> > Similarly, I do not think "data-augmented SFT" is a very convincing substitute for comparing to the Instruction Hierarchy.  It's harder to compare against the Instruction Hierarchy, as the code and models are not open.  The only option is to compare to GPT-4o-mini, which unfortunately introduces other confounders.  Therefore, I don't think we can expect you to compare to the Instruction Hierarchy.  So I think this means the statements about Instruction Hierarchy should be revised, and the paper should not try to insinuate that it has compared to Instruction Hierarchy by comparing to data-augmented SFT.
> >
> > I'm a bit puzzled by the statement that there is no loss of quality/utility compared to the base (undefended) model.  The author rebuttal says that this information is found in Figure 9 in the appendix, but I do not see that shown; Fig. 9 only shows results for SFT and PFT models, but not for the original base model before fine-tuning.  I was not able to provide this comparison for Llama anywhere in the appendix.  Perhaps the authors were thinking of Figure 10(a), which does provide this comparison for Gemma.  However it appears that there might be a massive loss of quality for the Gemma models, as the log-likelihood on Alpaca is very different for the base model vs the SFT/PFT models.  I am not sure whether the log-likelihood of Alpaca is a good measure of output quality/utility.  I suspect that accuracy on the password dataset is not a very sensitive/informative metric.
> >
> > I believe it is important to compare to the undefended model.  That is what consumers of these models will ultimately care about.  They will care less that your defense provides utility to some other defenses.  They will care a lot that your defense provides the same utility as current undefended models.
> >
> > I think "key instruction" remains a bit unclear.  I did read the part of the paper that the author response is referring to, but I still find it vague.  It appears there are some unstated premises/assumptions, e.g., that the system prompt contains one sentence that specifies the task and the other sentences are unnecessary/irrelevant/ secondary.  I find that questionable.  I bet in many cases the task is specified in multiple sentences.  I doubt that there is a clear separation between "key instruction" vs "the rest" in many/most system prompts found in practice.
> >
> > I view all of these comments as secondary, and fixable/addressable in a revision.  Thank you for the detailed, thoughtful, and thorough author response, and for sharing this research with me and the community.  I do find it valuable.

---

### Author Response · Authors · 2024-11-18
**Overall response**

We thank the reviewers for their comments.  We release our code and datasets (https://anonymous.4open.science/r/pft-iclr-08E2  ) for better reproducibility.

We want to clarify that (1) the goal of this paper is the scientific question: can we secure LLM with targeted SFT? If not, why? Then, to characterize this question and understand the phenomenon,  we conduct controlled experiments: more specifically, we (2) focus on closed-domain bots (3) study their robustness with respect to varying system prompt designs (in particular, varying location of the key instruction), measured by (4) generalizable defense against attacks. To answer the key research question, we also (5) tried our best to incorporate methods from the related work.

***(1) What this paper is about***

This paper is driven by a scientific question: can we secure the LLM through SFT? This question is too broad. So we start from the most simple setup (closed-domain LLM bots) and focus on one aspect of the model robustness: its sensitivity to the location of the key instruction. This allows us to discover interesting model behavior and gives a quantitative description of model vulnerabilities. These observations lead us to a hypothesis that the concatenated prompt format doesn’t clearly delineate different roles, and we provide remedies motivated by this hypothesis.


***(2) Focus on Closed-domain LLM bots***

As explained in the introduction (e.g. paragraph of line 39) we consider the closed-domain LLM bots where each bot solves one task (which we call the “key task”). In this setting, the developer specifies the key task with a “key instruction” inside the system prompt; ideally, the LLM treats the entire user input as data and applies the key task to it. In other words, the LLM should not follow any instructions in the user prompt.

Implementing LLM bots through system prompts is widely used (e.g. [GPT-store](https://openai.com/index/introducing-the-gpt-store/) , [Coze-store](https://www.coze.com/home) ), and even closed-domain bots have important applications like resume screening where the LLM must treat user input strictly as data rather than following embedded instructions.

**Why we didn't include open-domain applications** (i.e. allow model to follow “allowed” user instructions)? This adds complications that distract from the focused question: when the model follows an “unallowed” user instruction, it’s not clear if it’s because the model fails to distinguish between system and user role (which is our focus here), or that it fails to understand that such instruction is not allowed. So we can’t draw useful conclusions by including the open-domain settings.


***(3) Robustness w.r.t. System Prompt Designs***

Many papers have studied the model's robustness with varying prompt injection attacks.  However, in this paper, we ask another question: given a fixed set of prompt injection attacks, is the model robust to various system prompt designs? We quantitatively study the relationship between the model robustness and the location of the key instruction. Through this lens, we discover a surprising vulnerability — the LLM security actually depends on the location of the key instruction.

As argued in section 2.2, assessing the LLM sensitivity to various system prompt designs is important for real-world applications: at deployment time, system prompts are sometimes out-of-distribution from those used in security training; therefore we need to understand the model robustness at different system prompt designs.


***(4) Measure Generalizable Defense against attacks***

To assess model sensitivity to system prompt designs, we measure the generalizable defense against attacks. More specifically, we use only benign data during SFT, and evaluate using unseen attack data. This demonstrates the model's generalizable robustness against attacks.  Holding the training and evaluation data fixed, we perform controlled experiments that reveal model vulnerability, discover potential mechanisms, and evaluate remedies. While we do not test against more advanced attack data (Completion attacks, TAP, and GCG),  we argue the current setup is suitable for the purpose of this paper.


***(5) Comparing against other related work***
Our claim is not that PFT dominates all other defense methods. Rather, through PFT we demonstrate the importance of clearly delineating tokens between different roles. And we show it through controlled experiments: we use the same training and evaluation data, and the same algorithms; the only difference is the position-id manipulations.

We tried our best to include relevant components from the related work: our training data creation mirrors that in instruction-hierarchy for closed-domain applications, and we enhanced the delimiter in StrucQ as a baseline. Other parts of those works are cannot be incorporated, as they allow the model to follow "allowed" user instructions.

---

### Meta-Review · Area_Chair_jL9z · 2024-12-21

**Metareview:**

This paper proposes PFT, a novel position-enhanced fine-tuning approach that leverages position IDs to more effectively distinguish between system and user tokens. The strength of this paper are that (1) it studied an important question; (2) it presents a simple and elegant solution and (3) good performance. However, most reviewers provide the negative scores for this paper. Their concerns are (1) weak defense baselines (e.g., StruQ); (2) Weak attack baselines; (3) evaluation metrics metric; (4) practical usage; (5) insufficient experiments and (5) presentation problems. After rebuttal, although reviewer PTRg provides the postive score for this paper, all other reviewers still keep the negative score for this paper.  AC likes the simple but elegant solution of this paper. However, AC read them and agreed with reviewers' comments. Especially, although reviewer PTRg gives a positive score. But his/her further concerns (e.g., stronger prompt injection attack, baseline comparison) are still very important. AC feels the current paper is not ready for ICLR. But the idea is good. AC hopes the authors can revise the paper based on the comments.

**Additional Comments On Reviewer Discussion:**

In the discussion, the main concerns are still the stronger attacks and defense baselines. These concerns are not addressed well via rebuttal.

---

### Decision · Program_Chairs · 2025-01-22

Reject